# Variance in *C. elegans* gut bacterial load suggests complex host-microbe dynamics

Satya Spandana Boddu [1]*, K. Michael Martini[1,2], Ilya Nemenman [1,2,3], Nic M. Vega[1,2,3]*

**1** Department of Physics, Emory University, Atlanta, Georgia, United States of America, **2** Initiative for Theory and Modeling of Living Systems, Emory University, Atlanta, Georgia, United States of America, **3** Department of Biology, Emory University, Atlanta, Georgia, United States of America

* satya.spandana.boddu@emory.edu (SSB); nic.vega@emory.edu (NMV)

**Data availability statement:** All experimental data and analysis code used in this work is available at:
https://zenodo.org/records/14291768.

## Abstract

Variation in bacterial composition inside a host is a result of complex dynamics of microbial community assembly, but little is known about these dynamics. To deconstruct the factors that contribute to this variation, we used a combination of experimental and modeling approaches. We found that demographic stochasticity and stationary heterogeneity in the host carrying capacity or bacterial growth rate are insufficient to explain quantitatively the variation observed in our empirical data. Instead, we found that the data can be understood if the host-bacteria system can be viewed as stochastically switching between high and low growth rates phenotypes. This suggests the dynamics are significantly more complex than logistic growth used in canonical models of microbiome assembly. We develop mathematical models of this process that can explain various aspects of our data. We highlight the limitations of snapshot data in describing variation in host-associated communities and the importance of using time-series data along with mathematical models to understand microbial dynamics within a host.

## Author summary

Bacterial population density is known to vary across individual hosts. What drives this variation is unclear. In this study, we use *C. elegans* as an easily controllable host, controlling for the age of the host and genetic heterogeneity to address this question. We quantify populations of individual bacteria species and their interactions with *C. elegans*. We found that bacteria behaved differently when grown in a host compared to the standard logistic growth observed *in vitro*. When bacteria grew within the host *C. elegans*, they exhibited density-dependent growth and the emergence of two distinct subpopulations of worms, one with high bacterial density and the other with low bacterial density. We also observed that hosts can switch between high and low population densities of bacteria. To describe this behavior, we developed a phenomenological model and a switching model for the bacterial dynamics inside the host that we simulated and compared with experimental data.

**Funding:** This work was supported by the National Science Foundation (#2014173 to NV and IN). The funders had no role in study design, data collection and analysis, decision to publish, or preparation of the manuscript.

**Competing interests:** I have read the journal's policy and the authors of this manuscript have the following competing interests: NMV, who is a senior author on this work, is an editor for PLOS Computational Biology.

## Introduction

Microbiomes are complex, dynamic microbial communities [1–4], the composition of which is known to vary within and between hosts [5,6]. Many factors are thought to contribute to this variation, including: genotypic or phenotypic heterogeneity in hosts [7,8], diversity in host-microbe and microbe-microbe interactions [9,10] and stochasticity in the colonization process [11,12].

Gut microbial composition is known to vary across host species, across individuals in a population, and within individuals over time [13–15]. Canonically, this variation is described in terms of the diversity of bacterial taxa comprising a microbiome. However, increasing evidence suggests that variability in total abundance - particularly within individuals over time - may also be relevant for understanding these systems [14,16]. To better characterize variation in host-associated microbiome composition, we need to understand how different sources of variation contribute to the distributions observed in data.

The nematode *C. elegans* is a simple model host especially useful to study heterogeneity in microbial community assembly. The commonly used laboratory wild-type strain *C. elegans* N2 Bristol is androdiecious and reproduces primarily by self-fertilization, allowing production of highly homozygous populations under laboratory conditions. The short life cycle of this host facilitates generation of large numbers of age-synchronized adults on time scales convenient for laboratory experiments. This ability to produce large, highly genetically homogeneous, age-synchronized populations with a shared physical environment and life history allows for the generation of sufficient data to make the worm a mathematically tractable model for studying variation.

This host assembles a characteristic intestinal microbiome from soil bacteria in its native habitat [17,18]. For the gut colonization process, bacteria need to survive passage through the grinder in the worm's throat during ingestion and then attach themselves to the epithelial layer of the gut to grow inside the worm. The colonization process is stochastic [11], but composition of these communities is shaped by deterministic processes including interactions among microbes [9] and between microbes and the host [19,20]. As in other host-microbiome systems, there is substantial compositional variation in the gut microbiome among individuals. Interestingly, we observe variation even in populations of isogenic, synchronized hosts colonized from a shared inoculum in a uniform, well-mixed environment [20,21]. This result motivated us to investigate the drivers of community variation using the *C. elegans* model system.

In this paper, we used a combination of experimental data and mathematical modeling to understand what contributes to observed variation in microbial colonization within populations of hosts. In our previous work [20,22], where worms were colonized with an eight-member minimal native microbiome, we observed large variation in microbiome community composition and total bacterial load among individual hosts of the same genotype. In the current experiments, we further simplified this system by mono-colonizing synchronized populations of *C. elegans* N2 Bristol with each of the single bacterial isolates from the minimal native gut microbiome. We found that high variation is still present in bacterial load of mono-colonized worms, indicating that interspecies microbial interactions are not required to produce inter-individual differences in microbial colonization of the worm intestine. Using a logistic type neutral model to parameterize the colonization process for each bacterial isolate, we found that the observed variation is not well explained by demographic noise alone or by parametric differences among hosts. Instead, we found emergence of apparent alternative states in bacterial load inside worms, with transitions of individuals between these states, suggesting that canonical models of bacterial growth cannot fully characterize the host-microbe

system's dynamics. We find that the distributions observed can be qualitatively recapitulated by two distinct families of models, but that snapshot-type data of populations (instead of longitudinal time series of individuals) are insufficient to distinguish between these competing models, indicating the usefulness of autocorrelated data and absolute abundance measurements in understanding the dynamics of host-associated microbial populations.

## Results

### Large variation in total bacterial population size is observed in the host but not *in vitro*

We previously observed substantial variation in total bacterial load across individual worms colonized with a minimal native microbiome (species in Table 1) [20,22]. Here we sought to determine the sources of this variation. First, we sought to determine whether either the host environment or the microbial community context is required to produce such variation in distributions of bacterial population size. For this, we colonized wild-type N2 worms with individual bacterial isolates from the minimal native microbiome. Specifically, wild-type N2 worms were mono-colonized in liquid culture according to standard protocols [15], with each bacterial isolate presented at $10^8$ CFU/ml. Bacterial load in individual worms was quantified at specific time points (3…48 hours) after the start of colonization. For comparison, we also measured growth of individual bacterial isolates *in vitro* outside the host (as OD600) in a standard worm medium (liquid NGM).

   While growth *in vitro* showed low variation across replicates, we observed considerable variation in host-associated bacterial load between bacterial isolates and across individual hosts colonized with each isolate (Fig 1A). Different bacterial isolates grew to different final densities in both environments, but there was no obvious relationship between maximum density of a given bacteria *in vitro* and in the host. Standard deviations of bacterial load in hosts (colony forming units, CFUs, per host) scaled linearly with the mean in these data (see Fig A in S1 Text for data with other bacterial species). Consistent with previous work [23], distributions of log-bacterial load in hosts tended to be long-tailed and left-hand skewed (Fig 1B).

### Demographic noise does not explain variation in bacterial load

To verify if the variation across hosts mono-colonized with a given bacteria, indeed, cannot be attributed simply to demographic noise, we established a simple stochastic neutral model as our baseline model for these data. In this model, the only source of variation is demographic noise due to spontaneous colonization/birth/death events. This model contains three rate parameters, which are assumed to be identical across all hosts and are specific to a colonizing bacterial isolate. Colonization ($c_i$) corresponds to the process of bacteria entering the worm gut from the environment, death ($d_i$) corresponds to loss of bacteria within the gut, and birth ($b_i$) corresponds to bacterial multiplication within the host. Total occupancy of the worm gut is limited by a constant carrying capacity ($V_i$); $E$ corresponds to the number of empty sites in the worm gut (i. e., $E + N_i = V_i$ for each mono-colonized worm). Overall, the model is summarized as:

$$E \xrightarrow{c_i} N_i,$$

$$N_i + E \xrightarrow{b_i} 2N_i,$$

$$N_i \xrightarrow{d_i} E. \tag{1}$$

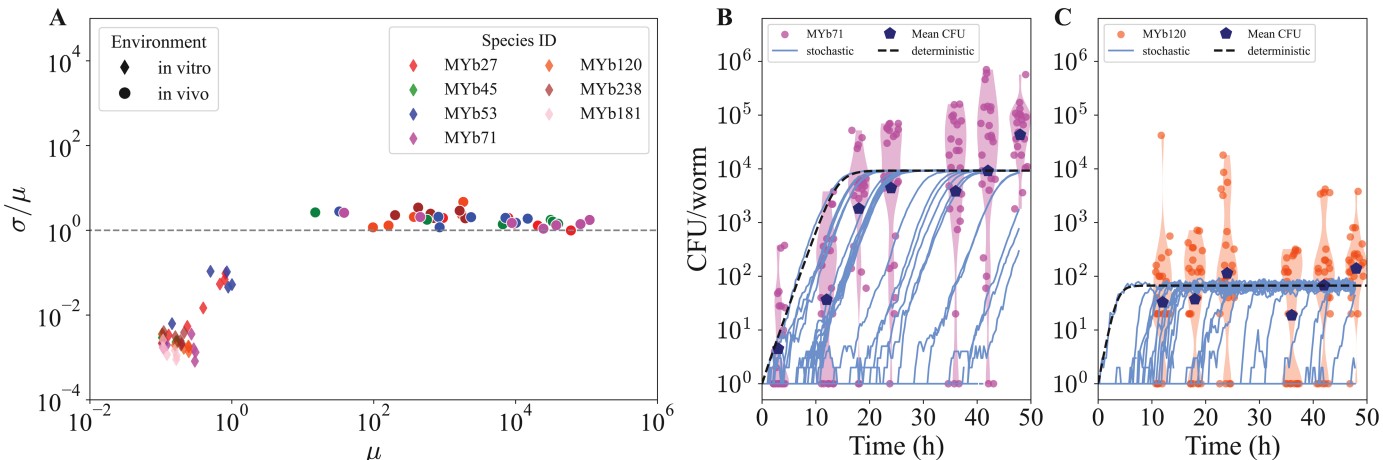

**Fig 1.** *In vivo* **bacterial populations exhibit high variation that is absent in** *in vitro* **populations.** A) Seven bacterial isolates from a *C. elegans* native microbiome (Table 1) were grown *in vitro* in NGM liquid medium at 25°C, measuring OD600 over 48 hours ($n = 6$ replicate wells). For *in vivo*, CFU/worm measurements were taken in individual wild-type N2 worms ($n = 24$) at time points 3…48 h during mono-colonization using the same set of bacteria. Comparison of standard deviation/mean to mean for both *in vitro* and *in vivo* across replicates or worms at different times is shown. (B-C) Bacterial load in samples of individual worms over time since the start of colonization. Experimental data are shown for one good colonizer (B, MYb71, *Ochrobactrum*) and one poor colonizer (C, MYb120, *Chryseobacterium*) to illustrate typical colonization dynamics; other bacterial species' data are shown in Fig A in S1 Text. Gillespie simulations (blue lines, $n = 30$) were carried out using parameters obtained from fitting CFU per worm data to the mean field model (Table A in S1 Text). Mean log-CFU at each time point (large blue point) and the mean-field deterministic simulation (dashed black line) are also shown.

This model is a single species special case of the model described in Martini et al. (2024) [22]. The mean-field equation for the model describes the mean population density, $\phi_i(t) = \left\langle \frac{N_i(t)}{V_i} \right\rangle$ as a function of time:

$$\frac{\partial \phi_i(t)}{\partial t} = (1 - \phi_i(t)) (b_i \phi_i(t) + c_i) - d_i \phi_i(t). \tag{2}$$

This model, in which bacteria grow according to a fixed set of parameters and all hosts are identical, provides a convenient null model for bacteria-host dynamics. If this model is sufficient to describe the variation in bacterial load that is observed within populations of worms, it implies that all worms are essentially identical to each other and share a fixed, time-and density-invariant set of parameters for host-bacteria interactions. Otherwise, one or more of these assumptions is incorrect and must be relaxed to find a sufficient model.

To test this, bacteria-specific parameters inferred by fitting the model (see Materials and methods: Overview of simulations and computational methods) to log transformed mono-colonization data (Table A in S1 Text for fitted parameter values) were used to initiate stochastic Gillespie simulations [24,25]. As expected, the model adequately captured the central tendency of the log data but fell short in explaining the observed variance, particularly at later time points, when convergence to saturation density was expected (Fig 1B and 1C, see also Fig A in S1 Text for results with other bacterial species). The parameterization of the model is not unique (birth and death partially compensate each other). However, this conclusion does not depend on the specific parameter set used. In our model, we consider the well-mixed volume inside the worm and the carrying capacity of the worm to be the same. For this reason, the fluctuations from our model are small in amplitude, which is inconsistent with the experimentally measured fluctuations. Thus, demographic noise alone was insufficient to account for differences in bacterial load between individual hosts.

## Static host heterogeneity does not explain the variation in bacterial load

One way of explaining the high variation observed in the bacterial load among individual hosts is to allow individual hosts to have different parameters for interactions with bacteria. This assumption has some biological support: there is a well-established "hidden heterogeneity" within synchronized, isogenic populations of *C. elegans*, which can be seen in distributions of aging, mortality, and stress response [26–30]. We therefore next hypothesized that parameter(s) of the logistic growth for intestinal bacteria might differ across individual hosts.

We first sought to identify a parameter or a parameter combination which, if allowed to be heterogeneous across individual hosts, could explain the observed data. We started with the ingestion and excretion rates. For this, we measured accumulation of bacteria-sized inert particles as a proxy for bacteria in individual worms and quantified the variability of this accumulation, see Materials and methods: Accumulation and loss of inert fluorescent particles. As seen in S1 Text: *Low variability in ingestion and excretion processes*, the measured variability was considerably smaller that for bacteria, Fig 1, suggesting that variation in the ingestion/colonization, $c_i$, and excretion/death parameters, $d_i$, cannot account for the observed bacterial load heterogeneity.

We next focused on variability of other model parameters, which are harder to probe experimentally, but whose effects can be observed via modeling.

**Heterogeneous carrying capacity.** We sought to determine if worm hosts significantly differ in their carrying capacity $V$. In a scenario where carrying capacity $V$ varies across hosts and all other parameters are the same, average time to first colonist and early (exponential) growth of colonists ($N$ far from $V$) should be the same regardless of the capacity of individual hosts. However, when migration rates ($c_i$) are changed experimentally, we expect at high migration rate to achieve the final distribution of carrying capacities earlier than when the migration rate is low. If the rate is increased in this scenario, waiting time to the first colonist during early colonization should decrease, resulting in fewer uncolonized individuals at early time points. The number of uncolonized individuals should decrease over time, also as a function of this rate. For a "strong" colonizing bacterial isolate capable of growing and maintaining its population inside the host, once an individual host is successfully colonized above some threshold of stochastic extinction, growth of bacteria in the intestine should proceed until carrying capacity is reached. For a "poor" colonizer where (birth-death) is small or negative, the migration rate should have a much larger effect on the observed bacterial load.

Previous work [11] indicated that the density of bacteria provided in liquid media can be used to control the average migration rate into the worm intestine. Thus we mono-colonized worms with one of two isolates (MYb71 or MYb120) across three orders of magnitude in inoculum density to determine the effects of colonization rate on distributions of bacterial load. Colonization over time (CFU per worm) was quantified via destructive sampling as previously described.

As shown in Fig 2, bacterial load varied across individual worms at all time points in all conditions. As expected, worms in the lowest-migration condition acquired colonists more slowly than worms in the highest-migration condition, with larger numbers of uncolonized worms at early time points. Distributions of bacterial load changed minimally between 24 and 36 h time points for both colonizing bacteria, suggesting that steady states had been reached.

For the "strong" colonizer MYb71, we expected to observe evidence of logistic bacterial growth within hosts. The data are broadly consistent with this expectation. Increasing colonization rate reduced the number of uncolonized individuals and up-shifted distributions of bacterial load in the earliest time point (12 h), whereas maximum bacterial loads at 24-36 h were similar across colonization conditions, suggesting growth of bacteria to maximum

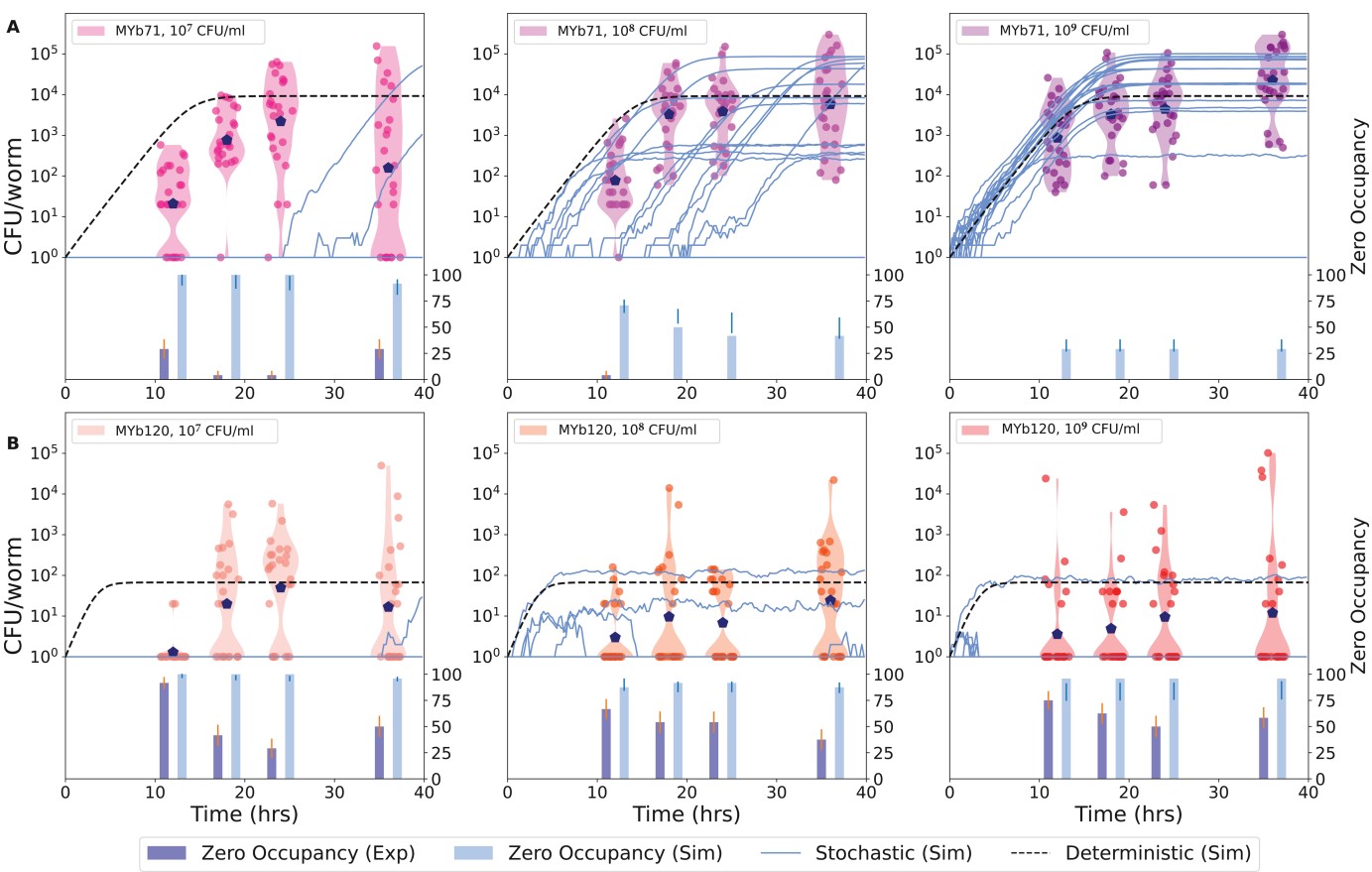

**Fig 2. A model with heterogeneous carrying capacity across worms is not sufficient to explain empirical variation in bacterial load.** Bacterial mono-colonization in wild type N2 worms by (A) MYb71 and (B) MYb120 as colonization rate is changed using bacterial densities in the inoculum: $10^7$ CFU/ml (left), $10^8$ CFU/ml (middle, same as Fig 1) and $10^9$ CFU/ml (right). Gillespie simulations (blue lines, $n = 24$) were carried out using birth and death parameters obtained from fitting single species log-CFU per worm data to the mean field model (Table A in S1 Text), and distributions of carrying capacity $V$ were established from data at 48 h post inoculation at the highest inoculation condition (far right). Colonization rates corresponding to the $10^8$ CFU/ml condition were taken from Table A in S1 Text and adjusted down and up by 10 fold in the lowest and highest colonization conditions respectively. Mean log-CFU at each time point (large blue point) and the mean-field deterministic solution (dashed black line) are also shown. Fraction of cases with the zero CFU at all time points is shown in bottom panels for experiments (dark blue bars) and simulations (light blue bars) (See Materials and methods for fitting and error bar calculations.)

capacity in the most heavily colonized hosts. However, bacterial load in the least-colonized individuals at 36 h was affected by colonization rate, with some individuals remaining below threshold of detection in the lowest-colonization condition.

For the "poor" colonizer MYb120, we expected a migration-forced scenario where bacterial load was determined by the balance between colonization and death in the intestine, such that increasing migration would increase the total load. This is not what we observed. Rather, while worms in the lowest-inoculum condition colonized more slowly, bacterial load was otherwise similar across conditions and time points. Recent work from another group suggests that this may be behavioral; MYb120 is mildly pathogenic, and worms exposed to high densities of a pathogen as a sole food source will stop feeding [31]. The result is that bacterial density in the inoculum has minimal effects on colonization, consistent with our data.

We next compared these results with predictions from a modified stochastic model, where individual hosts were allowed to have different carrying capacity $V$. Stochastic Gillespie simulations were initiated using the birth and death parameters from previous experiments (Fig 2,

Table A in S1 Text), colonization rate that is scaled to the bacterial density outside the worm ($c/10$ for $10^7$, $c$ for $10^8$ and $10c$ for $10^9$), and carrying capacity randomly picked from 36 h data of the highest bacterial density condition. The predicted trend of the means and first colonization in all the conditions agree with the observed data. Specifically, at 12 hours post colonization, the centers of the log transformed data move up with increase in colonization rate also agreeing with the deterministic prediction.

To compare simulations to the data, "zero occupancy" percentages (fraction of uncolonized worms) were calculated at each time point in each condition. For MYb71, the Gillespie simulations produced higher frequencies of zero occupancy (uncolonized worms) than observed in experiments under all conditions. For example, in the lowest condition ($10^7$) of MYb71 at the last time point (36 hours), we measured about 25% zeros in experiments, but the simulations predict about 75% zeros. This, in general, remained true for the poor colonizer MYb120: the expected and the observed fractions of uncolonized hosts were similar in the highest-migration condition and across conditions at the earliest time point, but simulations again over-predicted the fraction of uncolonized hosts at all later time points in moderate and low-migration conditions. For both colonists, fraction un-occupied worms did not consistently decrease monotonically in time, as would be expected from stochastic growth and colonization in a logistic framework. This suggests that the null model with heterogeneous carrying capacity in a population of hosts is insufficient to capture the variation observed in the experimental data.

**Heterogeneity in growth and colonization parameters.** To explain the variation observed in the experimental data, we next considered an extension of the previous model where the birth/death and colonization rates of bacteria in the host are allowed to vary between individual hosts. The intuition for this is that, whereas a 100-fold range in observed bacterial load would require a 100-fold range in $V$ if this parameter is allowed to vary across hosts, which is mechanistically difficult to justify, a relatively small amount of individual variation in the rate parameter(s) can accumulate over time produce the indicated range. Further, although colonization rate does not affect the final state of the system so long as $b-d>0$, and observed variation in input-output processes is not by itself sufficient to explain the data, individual variation in this parameter is expected to contribute to observed variation in bacterial load early in time. To determine whether individual variation in these parameters collectively was sufficient to explain the variation in the data, we conducted Gillespie simulations of bacteria colonization in worms with different birth ($b_i$) and colonization ($c_i$) parameters. The birth rate parameters of bacteria for a given worm are drawn from a normal distribution with a given mean birth rate and variance. Similarly, the colonization rate is drawn for each host from a normal distribution.

We found the best fit set of parameters (including the parameters characterizing the normal distribution of birth rate and colonization rate) that maximize the log-likelihood of data (see Materials and methods: Overview of simulations and computational methods). For the strong colonizer MYb71, we cannot reject this model as a way of describing the observed variation in bacterial load (Fig 3) . However, the model is not sufficient to explain the variation in the data for the poor colonizer MYb120. Additionally, we expect that if we were to increase the resolution of our log-likelihood calculation by increasing the number of levels of bacterial load (see Materials and methods: Overview of simulations and computational methods), the model fits would become worse for both colonizers as the predictions become more over-constrained.

**Heterogeneity in growth and death parameters under growth inhibition.** In our earlier experiments, total bacterial load consistently varied over orders of magnitude between

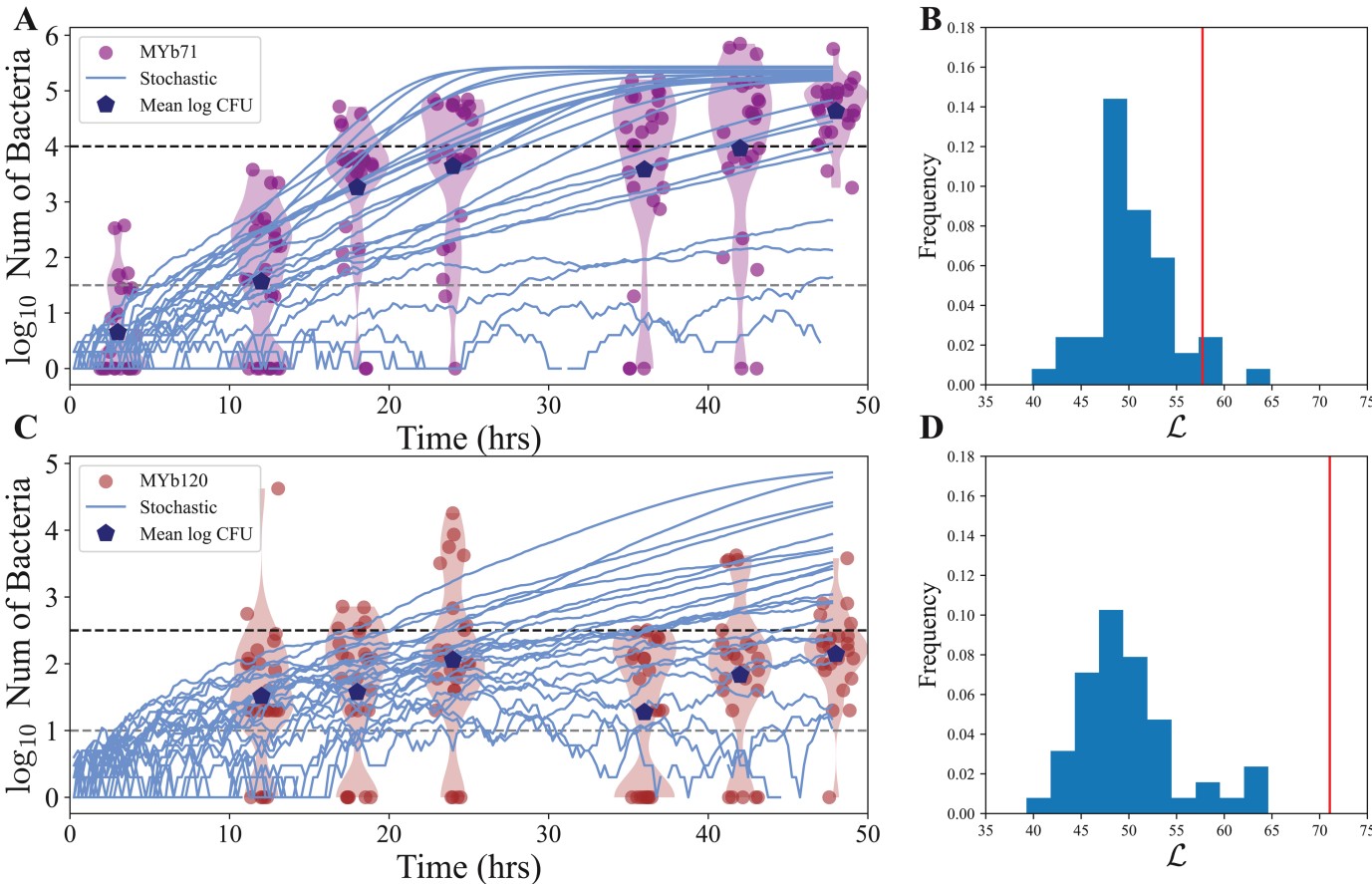

**Fig 3. Heterogeneous growth parameters generate large bacterial load variance, but the model is still insufficient.** Gillespie simulations (blue lines, *n* = 24) with heterogeneous growth parameters (colonization and birth rates) for A) MYb71 and C) MYb120 in comparison with the data previously shown in Fig 1. The upper bound of the low bacterial count bin and the lower bound of the high bin, which were used to determine the best fit parameters, are shown in dotted lines (see Materials and methods). Distribution of the negative log-likelihoods ($\mathcal{L}$) of the simulations (*n* = 50 with 24 simulations each) using the best parameters for the model for B) MYb71 and D) MYb120. Negative log-likelihood of the experiment (red line) given the best parameters is also shown.

individual hosts within synchronized, isogenic populations when colonized with single bacterial taxa under shared conditions [15,20–22]. However, the results thus far did not indicate that individual worms had different total capacity for bacterial colonization. Nor did we find evidence that individual worms had sufficiently different parameters for colonization or excretion rates (see S1 Text: *Low variability in ingestion and excretion processes*), and evidence for differences in birth/death rates was equivocal. It was, therefore, reasonable to ask experimentally whether there was support for differences in net bacterial growth (birth-death) among hosts. To clarify this, we next sought to isolate these within-host processes after an initial intestinal population was established.

As the host presents a responsive, biotic environment capable of controlling bacterial load, we hypothesized that different hosts might be differently able to exercise this control, and that this might correspond to differences in bacterial death rate in the intestine. Therefore, we modified our null model to allow bacteria-specific death rates inside the host to be heterogeneous across individual hosts. For simplicity, growth rates and carrying capacities in this model were bacteria-specific and constant across hosts within a population. In the absence of migration ($c_i = 0$) and at steady-state ($\frac{\partial \phi}{\partial t} = 0$, we can solve Eq 2 to get an effective steady state

value of $N_i^*$. In this model, high-load worms have bacterial birth rates much higher than death rates, resulting in an effective steady-state capacity for bacteria of species $i$ close to the true capacity $V_i$. If $b_i > d_i$, this capacity is:

$$N_i^* = V_i \frac{b_i - d_i}{b_i} = V_i \left(1 - \frac{d_i}{b_i}\right). \tag{3}$$

From this, simple qualitative predictions are possible for the effective steady state value $N_i^*$. If growth in the intestine $b_i$ were decreased while $d_i$ remained unchanged, effective steady state capacity should decrease according to the underlying rate (or population of rates) $d_i$.

To test this prediction, we used a bacteriostatic antibiotic to decrease growth of bacteria inside pre-colonized hosts. For these experiments, N2 adults were pre-colonized with fluorescently labeled *Ochrobactrum* MYb14, a member of the native worm microbiome closely related to the "strong" colonizer MYb71. Like its relative, MYb14 is a well-tolerated commensal and colonizes to high densities in the worm intestine (Figs 4, 5 and 6E). After pre-colonization, worms were exposed to a gradient of the static antibiotic chloramphenicol, including low drug concentrations sufficient to impair growth of MYb14 in the supernatant but not inside the host [32]. Heat-killed OP50 was used as an inert food source during antibiotic treatment. Pre-colonized worms were divided across antibiotic treatments at random, so that all treatment groups represented draws from the original population of worms. Worms were measured, washed, and re-fed with fresh inert food and antibiotic every 24 h. Confirming the utility of fluorescence as a proxy for bacterial load, the relationship between bacterial load and fluorescence in individual worms was monotonic (Materials and methods: Fig 7). Notably, this colonist showed a threshold of detection of roughly 100 CFU per worm, as the auto-fluorescence of *C. elegans* limits the ability to detect low bacterial loads inside the worm guts. Worms below the auto-fluorescence line in Fig 4 are not necessarily uncolonized worms as can be seen in Materials and methods: Fig 7. Further, fluorescence saturated at high bacterial densities (above $10^6$ CFU per worm), indicating a "ceiling" effect for highly colonized worms (Materials and methods: Fig 7). This resulted in a nonlinear mapping between the fluorescence signal and the CFU measurements, which needed to be addressed in the models below.

The lowest concentrations of drug used here (25 and 100 $\mu$g/mL chloramphenicol) are sufficient to inhibit bacterial growth outside the host, preventing new colonization, but do not appear to inhibit growth within the host. This is expected, as antibiotic concentrations required to inhibit growth in the *C. elegans* intestine are typically 5-10X higher than the *in vitro* MIC [32]. Accordingly, in both of these conditions we observed substantial growth of bacteria in the worm gut over 72 h; this was visible in total bacteria-associated fluorescence per host and in direct measurements of bacterial load (Fig 4, see also S1 Text: *Additional conditions for inhibition of bacterial growth in the intestine also show emergence of subpopulations with different bacterial loads* for illustration of variability in these measurements). At higher concentrations (250-500 $\mu$g/mL antibiotic), growth of bacteria within the host was partially inhibited, as can most easily be seen by comparing bacterial load after 72 hours on antibiotic (Fig 4).

Maximum intestinal populations increased over time overall even at the highest concentration of drug, indicating that growth was not reduced to 0 (Fig 4). This was observed in bacterial load taken from a sub-sample of worms as well as in GFP fluorescence, again indicating the utility of bacterial fluorescence as a proxy for bacterial populations in the host (Materials and methods: Fig 7).

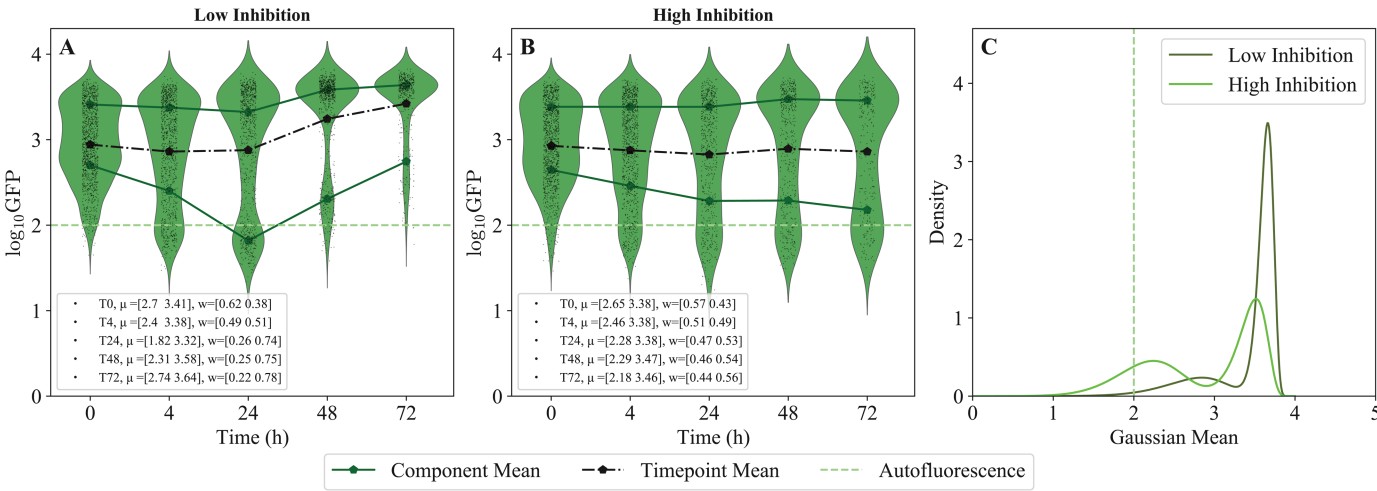

**Fig 4. Distributions of bacterial load under low vs. high inhibition of growth within hosts.** Green fluorescence in MYb14-GFP-KmR pre-colonized worms over 72 hours of inhibition at (A) low (25$\mu$g/ml) or (B) high (500$\mu$g/ml) concentrations of chloramphenicol. Centers and weights of high and low GFP modes at each time point from the transformed GMM fits are shown in the legend. Mean GFP (black dots, dashed lines) for the entire population at each time point is shown. Mean autofluorescence (light green, dashed lines) calculated using the data from uncolonized worms is also shown. CFU to GFP mapping is shown in Materials and methods: Fig 7. (C) Probability density fits to a Gaussian Mixture Model (GMM) with two components (Materials and methods: Overview of simulations and computational methods), fit on log(CFU) transformed data and then transformed back to log(GFP), at T72 for low and high inhibition (see Materials and methods: Calibration of log(CFU) to log(GFP) mapping).

However, differences between treatments indicated substantial growth suppression at the highest levels of drug — at 72 hours, median CFU/worm was 5.5 logs at low drug vs 3.8-4 logs at high drug. Under partial inhibition of growth within the host (high drug), by the final time point, we observed fewer worms at a very highly colonized state (5.5-6 log CFU/worm) and a larger fraction of individuals in the low colonization state (∼3.5 log) as compared to populations where only migration into the host was suppressed (low drug) (Materials and methods: Fig 7).

In the $c_i = 0$ regime, where migration is absent, the effective steady state is specifically given by Eq 3. From the heterogeneous models presented thus far, we expected to see steady state values near the carrying capacity, as $d \ll b$ for this bacterial colonist. When bacterial growth is reduced by antibiotic treatment, we would expect that the steady state bacterial load would be reduced accordingly. However, we do not observe this (Fig 4). This could be due to the fact that the death rate is much smaller than the birth rate which would result in $N_i = V_i$ and we cannot observe the small downward shift due to biological variance masking this shift.

Further, these data indicated multiple modes of bacterial density within populations of hosts. We fit these modes to the Gaussian Mixture Model (GMM) with two components, as detailed in Materials and methods: Overview of simulations and computational methods. The weight of each mode corresponds to the fraction of the worms in the corresponding bacterial load state (Fig 8). For example, in the high drug condition at 72 hours, the weights of the two modes ("high" and "low" bacterial loads) are comparable. While the GMM fits are better interpreted as a summary statistic than as a complete partition of data, there are clear signatures of multiple subpopulations in our data. Furthermore, changes in population weight (fraction of worms in each of the modes) across modes of the GFP fluorescence data cannot be explained by sampling effects; the entire population of worms is measured at each time point, and all of the same worms (minus losses to handling) are present across time points, so this effect is unlikely to be an artifact due to sampling. One possible hypothesis is that there

could be, indeed, two sub-populations of worms, where intrinsic birthrate ($b$–$d$) is either high or low. If this were true, we would expect to see bacterial load in the "high" sub-population to stay high over time, and vice versa for worms in the "low" sub-population. However, in Fig 4A, we also observed that the "low" subpopulation at 24 hours does not continue to stay low; the re-distribution of weights suggests some net movement of individuals from the "low" subpopulation to the "high" subpopulation over time. Although it is not possible to determine trajectories of individual worms from these data, it is difficult to explain the observed redistribution of population weights across the modes without some corresponding transitioning of individual worms between them.

## Data suggest multiple worm subpopulations, with individuals transitioning between them on scales of hours

We, therefore, next sought to isolate worms that differed in initial bacterial load, to determine whether the redistribution of individuals seen in Fig 4 across states implied by these data was actually occurring.

From the logistic model, we expected that if net growth of bacteria in the host was different in high- and low-bacterial load worms, these groups would respond differently when an initially high rate of colonization was reduced to 0 after establishment of a bacterial population. High-load worms, where presumably initial birth is much larger than death during early colonization, should be well out of the migration-forced regime (where bacterial load is heavily dependent on migration) and should be minimally responsive to any change in colonization after the initial population is acquired. Low-load worms, where presumably these rates are more similar, may drop slightly in bacterial load or even decrease toward extinction (if death is larger than birth, such that bacterial load in these worms is entirely dependent on migration). Removing colonization should, therefore, increase separation between modes, with high-load individuals remaining highly colonized and low-load individuals settling at a level determined by the relative contributions of death and migration.

To establish initial populations, N2 adults were pre-colonized for 24 h on fluorescently labeled *Ochrobactrum* MYb14. After pre-colonization, worms were sorted into "high" and "low" GFP bins based on total fluorescence, with a narrow gap between bins to minimize overlap in bacterial load. (Note that these broad bins were expected to have considerable internal variation, from the machine or across experiments.) Worms were then split evenly between treatments, with half of the worms from each bin returning to the original condition ($10^8$ CFU/mL live MYb14-GFP provided in the media, denoted below as "Migration"), and half moving to a condition where new colonization is prevented (heat-killed OP50 for inert food + chloramphenicol to prevent cross-inoculation among hosts, denoted below "No Migration"). Populations were measured at 24 and 48 h, so that each data set represents repeated measurement of the same population of individuals (minus individuals lost during handling).

We observed that individuals could transition out of their original bins, moving from "high" to "low" and vice versa. For example, in the absence of continuous colonization (no migration), worms starting from the "high" condition produce a "low" mode (Fig 5A) and vice versa (Fig 5B). Allowing continuous colonization (migration) (Fig 5D and 5E) changed the distribution of individuals but did not prevent moving between bins. This can be seen in distributions of GFP fluorescence (Fig 5, right most column) as well as in CFU/worm measurements from destructive sampling (Materials and methods: Calibration of log(CFU) to log(GFP) mapping). Similar to before, worms below the auto-fluorescence line could have low bacterial loads instead of true zeros. We observed a similar pattern in N2 hosts colonized

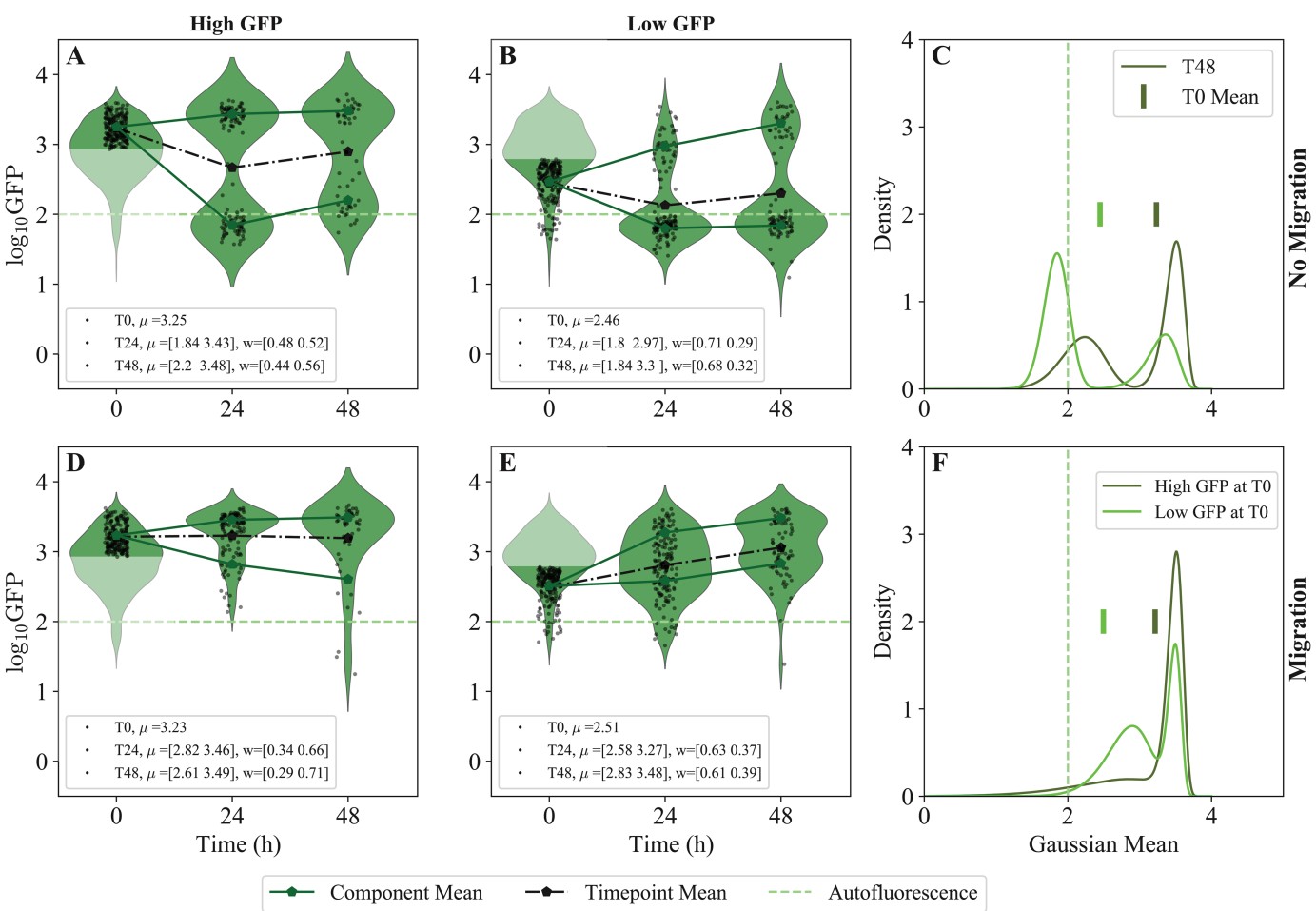

**Fig 5. Worm switch between high and low colonization states.** Green fluorescence over 48 hours after MYb14-GFP-KmR pre-colonized worms were separated based on high (A,D) and low (B,E) GFP under conditions of A-C) No Migration (top) or E-F) Migration (bottom). Full T0 distribution (pale green) in fluorescence is shown in the background. Mean GFP (black dots, dash-dotted lines) for the entire population at each time point is shown. Mean auto-fluorescence (Light green, dashed lines) calculated using the data from uncolonized worms is also shown. C,F) PDFs using the transformed GMM fits at T48 for worms starting at low (light green) and high (dark green) fluorescence in no migration (top) and migration (bottom) conditions (S1 Text: *Sample fits of models of bacterial load to GFP measurements*).

with a different bacteria, the pathogen *Salmonella enterica LT2* (S1 Text: *State switching is also observed with other bacteria*). Existence of two host subpopulations and individuals transitioning between them on scales of just a few hours is not consistent with the simple logistic model or with any host-sub-population model considered thus far. Instead, this indicates the need for a model capable of producing alternate states within individual hosts.

## Modeling multiple states in the worm-bacteria system

There are several possible ways to achieve multi-stable populations of bacteria within the intestine of *C. elegans*. Here we will examine two of them. The first model considers a system where multistability arises due to density-dependent population dynamics within the host, such that each stable population of bacteria has a basin of attraction characterized by a specific mean value. We can model this kind of multi-stability using multiwell-potential models from classical mechanics, where the potential ($U(\phi)$) contains multiple troughs

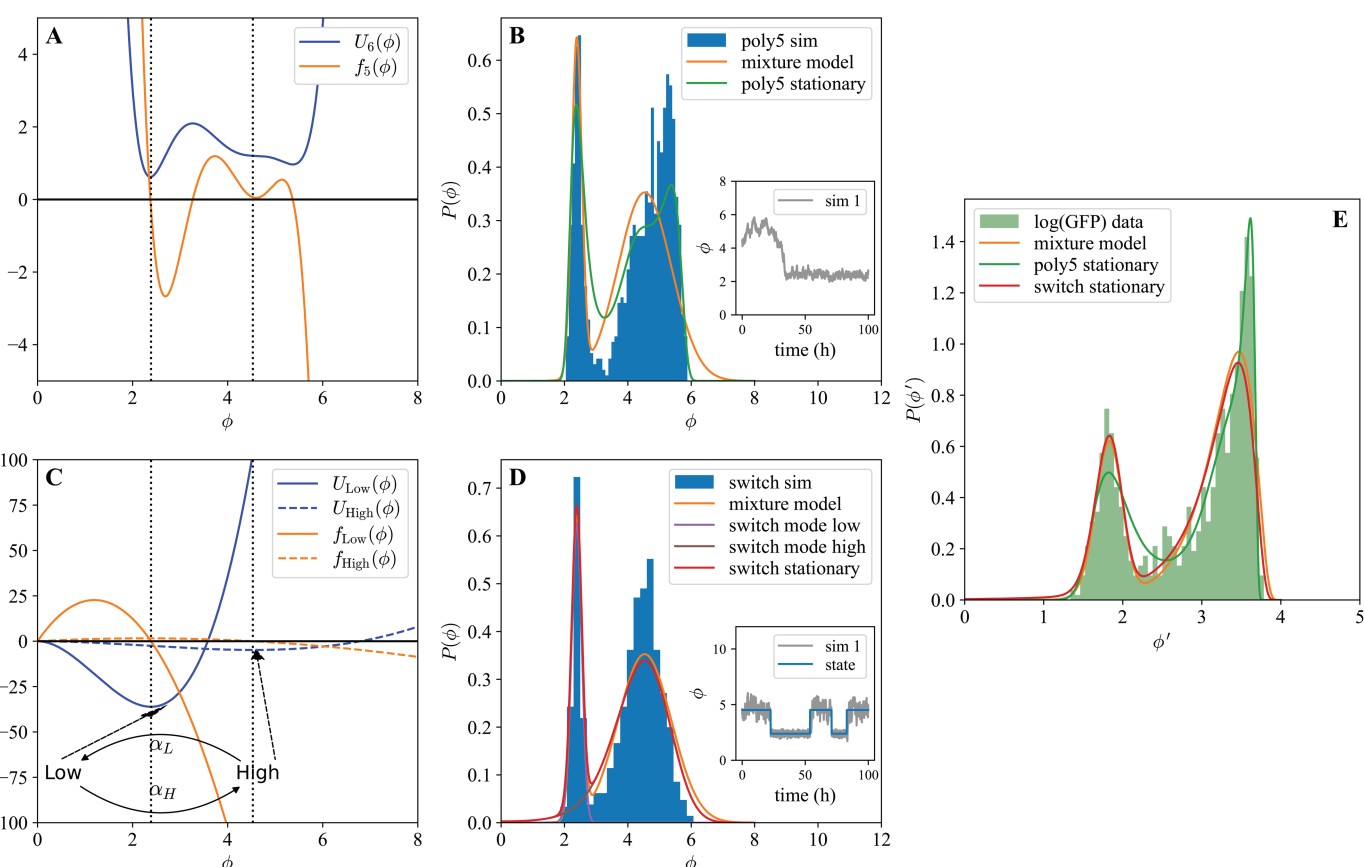

**Fig 6. Multi-stable population models.** A) Potential $U(\phi)$ and $f(\phi)$ for potential model fit to 24h low inhibition data from Fig 4 B) Probability distribution from 1000 simulations of the potential model, as well as the model's stationary state. Label poly5 corresponds to the fifth order force $f_5$ in panel (A). We also show the GMM fit for comparison. Inset shows one sample simulation. C) Schematics of the state switching model switching between Low and High states and the corresponding logistic growth functions $f$ for worms in the low ($f_{low}$) and the high ($f_{high}$) state and their corresponding potentials, $U_{low}$ and $U_{high}$. D) Probability distribution from 1000 simulations of the switching model as well as the model's low and high state distribution and the distribution in the stationary state. We also show the GMM fit for comparison. Inset shows a sample simulation and its corresponding state (high or low). E) Probability distribution of log(GFP) ($\phi'$) data and best fit theory curves (all transformed to the log(GFP) state) for the GMM, the sixth order potential model, and the state switching model.

corresponding to stable states in the population of the bacteria. Transitions between these states are driven by random white noise $\eta$, with zero mean $\langle\eta\rangle = 0$, and variance $\langle\eta(t)\eta(t')\rangle = D\delta(t,t')$. This model has the following expression:

$$\frac{\partial\phi}{\partial t} = f(\phi) + \eta = -\frac{\partial U(\phi)}{\partial\phi} + \eta \tag{4}$$

where $f(\phi)$ has roots at the fixed points, or alternately, the potential $U(\phi)$ has troughs (stable fixed points) and peaks (unstable fixed points). The density-dependent dynamics can be due to various reasons, which we are agnostic to in our model. In particular, these reasons can include intra-bacteria interactions.

We chose to fit the multiwell-potential model to the modes of the 24-hour time point in the low-inhibition data set in Fig 4. This time point is 72 hours after the *C. elegans* were initially colonized and is, therefore, on the same day of adulthood as the 48 hour time point of the no migration condition of Fig 5. This will allow us to avoid fitting transient behavior and

to simulate and compare to the no migration condition of Fig 5 (see Materials and methods: Probability distributions in the potential model).

Log(GFP) (denoted as $\phi'$) is a nonlinear monotonic function of the underlying bacterial population size inside the gut of *C. elegans*. It also depends on properties of the measuring device. This poses a challenge for modeling as the transformation between the fluorescence and the bacterial load is not obvious and must be fitted. To build a more reliable model, we transform the log(GFP) measurements to the log(CFU), denoted as $\phi$ a more accurate measurement of the bacterial load, using a nonlinear transformation inferred in Materials and methods: Calibration of log(CFU) to log(GFP) mapping. Fitting data in the log space with the potential model and white noise is equivalent to a multiplicative noise model in real space. Fluctuations are symmetric in log space, simplifying fitting procedures (S1 Text: *Modeling in the log(CFU) space results in a multiplicative noise process*).

The transformed data shows two large modes in the log(CFU) bacterial load $\phi$, which correspond to two troughs in the potential (Fig 6A). To have the two modes, the potential $U(\phi)$ must be at least 4th order or higher; or equivalently, that $f(\phi)$ must be 3rd order or higher. However, the fits of the potential using a procedure outlined in the Materials and methods: Probability distributions in the potential model showed that, while the 4th order potential is able to capture the means of the two modes of the bacterial load distribution, it struggles to capture the widths of the two main peaks Materials and methods: Fig 9). In contrast, the 6th order potential (equivalently, 5th order $f$), can do both. A plot of the simulation of the 6th order potential model is compared to data in Fig 6B. Crucially, even though the fits are done only using the snapshot data, the model also produces the dynamics of switching that are qualitatively similar (but not identical) to those seen in the experiments. In particular, using parameters determined from the 24 hour mark of Fig 4 to predict the no migration condition of Fig 5 we see in Fig G in S1 Text that the locations of the two main peaks are well predicted but the bacterial loads are not. Qualitatively the relaxation to the stationary distribution is similar between experiment and simulation. For example, when initialized in the high peak, some of the bacterial load transitions to the low peak with time. We speculate that the qualitative shape of the potential could come from the population density-dependent birth rates or death rates. These rates themselves can come from interactions between bacteria of the same species. The analysis in S1 Text: *Mathematical models struggle to predict quantitative details of the dynamics of the bacterial load* also shows that our model of the switching process is too simple, not allowing for quantitative prediction of the switching dynamics; more data are needed to hone onto the correct model of the switching process.

The second type of model is one where each *C. elegans*-microbiome system switches between states. We consider a model with two states, corresponding to "high" and "low" bacterial loads. We speculate that these two states could be due to something as simple as whether the *C. elegans* is feeding or not. It could also be some physical change in the gut, the mucosa, or the spatial structure of the bacteria in the gut. We denote the state variable as $s$, and it can take two values $\{s_{\text{low}}, s_{\text{high}}\}$. The system randomly switches between the states with $\alpha_H$ being the rate of low-to-high transition, and $\alpha_L$ the rate of the high-to-low one (Fig 6C). In each state, the bacteria grow logistically within the host intestine according to state-dependent parameters (Fig 6C).

$$\frac{\partial \phi}{\partial t} = r_s \phi (C_s - \phi) + \eta(s), \tag{5}$$

where $\langle \eta \rangle = 0$, $\langle \eta(t)\eta(t') \rangle = D_s \delta(t, t')$.

For comparison to the first model, the carrying capacity in the high state ($C_{High}$) corresponds to the highest root of *f* and the carrying capacity of the low state ($C_{low}$) corresponds to the lowest root of *f*. The rates of transition between states can be independent of the potential function describing each state. This is in contrast to the multi-well potential model where the curvature of the potential and thus the variance of the fluctuations contribute to controlling the switching rate between wells. The state switching model results in a probability distribution very close to that of a GMM (Fig 6D), making it straightforward to fit based on the estimated parameter of the GMM. Additionally, we are able to directly measure the switching rates between states and find the transition from low to high to be $\alpha_h = 0.08$ (1/h), and from high to low to be $\alpha_l = 0.02$ (1/h) (see Materials and methods: Probability distributions in the state switching model). Crucially, just like for the potential model, S1 Text: *Mathematical models struggle to predict quantitative details of the dynamics of the bacterial load* shows that the state switching model cannot produce quantitative fits to the dynamics of switching data; again, more data is needed to build a more accurate model of the underlying process.

A comparison of the GMM, the potential model, and the switching model is shown in Fig 6E. All the models were fit in log(CFU) space and then transformed back to the log(GFP) space for direct comparison to experiments. The GMM and the switching model are functionally identical in terms of their ability to fit the data. This is because each peak can be fit independent of the other. Both of these models struggle, however, to fit the right most part of the observed distribution. The potential model is better capable of fitting the right most part of the distribution of bacterial loads, but at low order it struggles to quickly transition between peaks. Clearly, as the order of the potential model is increased it becomes better able to fit the empirical bacterial load distribution.

While the two models are very similar, they do make some distinct predictions. Thus, in principle, experimental data can be used to detect which of the models is a better representation of the reality. However, in practice, the data we have is not statistically powerful enough for this purpose. For example, the independence of state transition rates and local fluctuations is one of the key predictions of the state switching model compared to the high-order potential model, where these features are dependent. However, to detect if the local fluctuations and the state transitions are independent or not with high accuracy requires single-host time series with sufficient temporal resolution to measure both features. For our population-wide "snapshot" data, these models are functionally equivalent as long as the potential is of high enough order. Indeed, we can see this from simulations of how these models behave when initialized as in Fig 5: both display qualitatively similar features as they approach their long time-steady state behavior (see Fig G and Fig H in S1 Text).

In other words, our data are sufficient to establish the existence of the two subpopulations of worm-bacteria systems, and the switching between the subpopulations on the scale of hours. However, the data are insufficient to either model the nature of the switching process, or the precise dynamics of the switching.

## Discussion

In this work, we show that there exists innate heterogeneity in colonization density within a population of individual hosts, with total bacterial load consistently varying over more than an order of magnitude between individuals. This heterogeneity was not apparent in *in vitro* bacterial populations, consistent with the expectation that bacteria grow *in vitro* in a approximately logistic manner and that cultures inoculated simultaneously from a single parent culture should exhibit low variation [33,34]. This observation suggested an important

role for the host, but the data were not well described by simple stochastic logistic models of host colonization. First, using Gillespie simulations assuming all the parameters involved in colonization are the same for all worms, we show that the demographic noise is insufficient to explain the variation we observe in our experimental data. Alternate models of the same form but with a stationary worm-to-worm heterogeneity in various population dynamics parameters were also insufficient to explain the data. Further, when new colonization was stopped after establishment of the intestinal population, the data indicated two distinct modes in the distribution of bacterial load across worms, with apparent transitioning of individual worms between these modes.

As these worms were sterile at the point of mono-colonization, inter-species bacterial interactions are not present; however, worms were raised on *E. coli* as a food source, and it is always possible that rearing conditions will differentially affect the physiology and/or behavior of the host. We have not extensively investigated the effects of different rearing conditions on microbial load dynamics. However, it is worth mentioning in this context that wide variability in microbial load is observed across a substantial range of experimental conditions (different host genotypes; feeding on plates vs in liquid; different rearing temperatures; different single-species colonists; microbial communities) [23].

We concluded that a canonical logistic type model was not sufficient to explain the population dynamics in the microbiome in the worm-microbe system, and that a suitable minimal model must allow multiple stable states even during mono-colonization of the host. Here we explored two such models: a state-switching model where the parameters of logistic growth are determined by a state variable, and a multiwell-potential model, where bacterial growth within the host is a higher order polynomial function of bacterial density. We show that both of these models are capable of reproducing the salient features of the bimodal distribution of the colonization data at different time points. Furthermore, while the models are distinguishable in principle, we cannot distinguish between them in practice with the data available from our experiments. Such disambiguation would require time series of bacterial load in individual worms, rather than dynamics of the distributions of the load over the worms, which we collect. These data can be collected using fluorescence microscopy with labeled bacterial strains [35–37]. A sufficiently granular time series should show whether changes in bacterial load behave more as expected for dynamical state transitions (movement between steady states on a more or less fixed landscape) or for an underlying system-level state switch (such that one primary state is available for each host-microbiome system at a given time, and the available state(s) change depending on some hidden process).

Although we found that the logistic model did not allow the dynamics observed in these host-associated microbial populations, the stochastic logistic model (SLM) framework has been used successfully elsewhere to recapitulate a number of empirical macroecological patterns in microbial communities [38,39], suggesting the general utility of the SLM as a null model for these patterns. Further, an extension of the SLM with alternate states — characterized by differences in carrying capacity for specific taxa, with transitions within individual hosts as well as differences between individuals — has been proposed to describe trends in longitudinal human gut microbiome data [40]. A similar observation emerges from quantitative profiling of gut microbiome composition over time [41], with the additional empirical insight that within-individual variation is less evident in relative than in absolute abundance data. Our results are consistent with the idea that state transitions are important for understanding the dynamics of host-associated microbiota, and the observation that these transitions exist in mono-colonized hosts indicates that a minimal model should have these transitions as a property of host-microbe systems, rather than a product of microbe-microbe interactions. Further work is necessary to define a sufficient (and hopefully unique) minimal

model for host-microbe dynamics and to understand the relationship between the dynamics of these systems and the patterns that emerge from this underlying ecology.

The drivers of bacterial load variation and "switching" in this system remain to be determined. In these experiments, we used isogenic, age-synchronized populations of hosts with a shared life history; genetic and developmental variation between individual hosts is minimal. While it is well known that worms within isogenic, synchronized populations are physiologically heterogeneous [26,28–30], we found that "hidden" host-side heterogeneity was not sufficient to explain these data. The range of bacterial loads per individual host [20,21] and the kinetics of bacterial accumulation over time (Fig 1, see also Fig A in S1 Text) vary across different colonizing bacteria, and for a given bacteria or community across worm genotypes, but these properties are generally consistent within a combination of host genotype and microbial colonist(s). This suggests that variation in the host-microbe system cannot be explained by variation in its constituent parts (worm alone or microbe alone). The more complicated, multistate models considered here are best interpreted as representing different forms of feedback within the system, where the available states in bacterial load are some function of the bacterial load (possibly as interpreted by the host immune response, by changes in intestinal function, etc.) or manifestations of an underlying system state (in the simplest case, from a history-independent process such as a stochastic switch). However, the data we have is not powerful enough to model the switching process in detail, or even to distinguish if the class of the model that explicitly involves multiple system states is a better representation of the process than the class of models with a population-size dependent growth and death. While an increase in the data set size would be beneficial, we anticipate that even large datasets of the type we have collected will not be able to fully characterize the transition process and propose biologically verifiable hypotheses about its nature. Instead, a different kind of data— longitudinal data on individual hosts—would be more powerful. Further, as the host appears to play a crucial role in controlling bacterial populations, and probably in transitions between states, it is plausible that an examination of host transcriptional and/or translational responses within different states would shed some light on plausible mechanisms of the observed multimodal bacterial populations. Both of these are goals for future work.

## Materials and methods

### Bacterial strains

Bacterial strains shown in Table 1 were inoculated from glycerol stocks and individually grown for 48 hours in 1ml LB cultures at 25°C with shaking at 300 rpm. All bacterial strains are from the MYb collection of native *C. elegans* gut isolates maintained by the Schulenburg lab [17]. Strain MYb56 (*Bacillus*) from the minimal native microbiome used in [20] was excluded from these experiments due to its spreading colony morphology on solid agar,

Table 1. Bacterial strains used in our experiments.

| Strain ID | Bacterial species |
| --- | --- |
| MYb14 | *Ochrobactrum* sp. strain BS30 |
| MYb27 | *Arthrobacter aurescens* |
| MYb45 | *Microbacterium oxydans* |
| MYb53 | *Rhodococcus erythropolis* PR4 |
| MYb71 | *Ochrobactrum pecoris* |
| MYb120 | *Chryseobacterium* sp. strain CHNTR56 |
| MYb181 | *Sphingobacterium faecium* |
| MYb238 | *Stenotrophomonas* sp. |

which limited the precision of counts. The fluorescent strain MYb14-KmR-GFP, bearing the broad host range plasmid pBTK519-KmR-GFP, was used for measurement of bacterial load via green fluorescence [42].

## Worm maintenance

Worm strain N2 was obtained from *Caenorhabditis* Genetic Center, which is funded by NIH Office of Research Infrastructure Programs (P40 OD010440). Standard protocols were used to maintain and cultivate worms for experiments [43]. Briefly, worms were maintained at 25°C on NGM plates with OP50 and synchronized using Bleach/NaOH protocols. Eggs were kept in M9 worm buffer overnight at 25°C with shaking at 200 RPM to hatch. L1 worms were then washed thoroughly in M9 buffer containing 0.1% Triton-X (henceforth M9TX01) and moved to NGM plates with *E. coli pos-1* RNA interference (RNAi) to produce reproductively sterile adults. After 60-72 hours at 25°C, reproductively sterile adult worms were moved to liquid S medium with 2X heat-killed OP50 (HKOP50), 200$\mu$g/ml gentamycin, and 100$\mu$g/ml chloramphenicol for 48 hours (25°C with shaking at 200 RPM) to remove any live bacteria that may have accumulated in the gut during growth. Germ-free adults were then used for bacterial colonization experiments.

## Single species colonization

Bacteria were grown individually to prepare inocula for mono-colonization of worms. Cultures were diluted to $10^8$ CFU/ml by suspension in S medium + 1% AXN (axenic medium), centrifuging them for 2 minutes at 10000 rpm and resuspension in the same medium.

Germ-free adult worms were sucrose washed prior to bacterial colonization to remove heat-killed bacteria, dead worms, and other debris. After sucrose wash, 100 worms per colonizing bacterium and time point were sorted into individual wells of a 96-well plate using a large object sorter (BioSorter, Union Biometrica). Each set of worms was washed 2-3 times with 1ml M9TX01, then suspended in 100 $\mu$l S medium + 1% AXN and 100 $\mu$l $10^8$ CFU/ml of individually-grown bacteria from Table 1. At indicated time points (3, 12, 18, 24, 36, 42, 48 hours), worms were removed from wells and washed to remove the bulk of external bacteria, halting colonization. Pre-colonized worms were incubated in S medium + 2X HKOP50 for 30-60 minutes to purge non-adhered bacteria, then rinsed 2X with 1 mL M9TX01 to remove OP50 biomass.

## Single worm digests

Single-worm digests were carried out according to standard protocols for this laboratory [15]. Briefly, after purging on HKOP50, pre-colonized worms were chilled to halt peristalsis and lightly surface bleached (1:2000 v/v) in cold M9 worm buffer on ice for 20 minutes to kill any remaining external bacteria. Worms were then rinsed 3X in cold M9TX01 to remove bleach and sorted individually into wells of an Axygen 96-well plate pre-filled with a small quantity of silicon carbide grit and 198 $\mu$l M9TX01 for mechanical disruption. Plates were chilled for one hour at 4°C, then sealed with a layer of parafilm and an Axygen square-well sealing lid to create a water-tight seal on individual wells. Plates were then shaken on a TissueLyser II to mechanically break up worm tissues (90 seconds at 30 Hz; rotate plates 180°; 60 seconds at 30 hz). After mechanical disruption, released gut contents were serially diluted in 1X PBS and plated onto salt-free nutrient agar (per L: 3g yeast extract, 5g tryptone, 15g agar), on which different bacterial colony morphologies can be easily distinguished. After 48 hours at 25°C, colony counts were recorded.

### BioSorter - green fluorescence experiments

**Inhibition.** Worms were pre-colonized on $10^9$ CFU/ml of MYb14-GFP-KmR for 48 hours as in Materials and methods: Single species colonization. Upon sorting, worms were suspended in S medium + 2X heat-killed OP50 along with varying concentrations of chloramphenicol (25-500 $\mu$g/ml). At selected time points (6, 24, 48, and 72 hours), green fluorescence was recorded across all conditions. At selected time points, sub-samples of worms were separated for digestion and plating in order to map CFU counts to the recorded green fluorescence signal in individual worms.

**State separation.** Worms were pre-colonized on $10^9$ CFU/ml of MYb14-GFP-KmR for 24 hours as in Materials and methods: Single species colonization. A small sub-sample of pre-colonized worms (n=100-200) were measured on BioSorter to establish gate boundaries for "high" and "low" fluorescence bins. The measured Green range from the BioSorter for "high" was around 850–3950 and for "low", it was 45–620 (arbitrary units). The bulk remaining worms were then sorted into high and low bins based on the measured fluorescence signals ($n$ = 150). Worms from each bin were then moved into each of two conditions: one where the original density of live fluorescent bacteria was provided, and one where worms were moved to an inert food source (S medium + 2X HKOP50 + 25 $\mu$g/ml chloramphenicol to prevent bacterial growth and re-inoculation). At 24 and 48 hours, worms were rinsed with M9TX01 to remove bacterial biomass, and green fluorescence was recorded on BioSorter for all individuals in the population. At 24 hours, all adult worms were retrieved after measurement (>99 % sorting retrieval efficiency), lightly surface bleached to prevent transfer of any contamination from the sorter, then returned to the originating condition until 48 hours. At 48 hours, green fluorescence was recorded on BioSorter, and a sub-sample of individual worms ($n$ = 24 – 36 per condition) were sorted into individual wells for disruption as in Materials and methods: Single worm digests.

### Accumulation and loss of inert fluorescent particles

In these experiments, germ-free adult N2 worms were prepared according to standard protocols, then exposed to fixed concentrations of Fluoro-Max green fluorescent polymer microspheres (0.21 $\mu$m diameter, particle density in aqueous solution 1.05 $g/cm^3$, cat. G200) in M9 worm buffer + 0.01% Triton X-100 to help prevent clumping of microspheres. See S1 Text: *Low variability in ingestion and excretion processes* for additional details [47].

### Calibration of log(CFU) to log(GFP) mapping

We expect the relationship between the concentration of bacteria and the amount of fluorescence detected through the BioSorter machine to be nonlinear. In particular, we expect this to be important near the saturation of the machine, which compresses a large range of bacterial loads (measured by CFU) into a smaller range of reported optical GFP readout. This has a consequence of introducing a skew in the GFP space, as the right side of the distribution is more compressed than the left.

To quantify how to transform the GFP readouts into the equivalent CFU measurements, wild type worms were colonized with MYb14-GFP-KmR for 24 hours as before, then run through BioSorter to measure the GFP fluorescence. During the sorter run, individual worms were sorted out for quantification of bacterial load according to standard protocols for this lab [15]. Plotting of GFP fluorescence vs bacterial load for individual adult worms indicated an approximately monotonic GFP to CFU mapping (Fig 7C). The log CFU to log GFP mapping was fit to a hyperbolic tangent function of the form, which allows for linear behavior at small

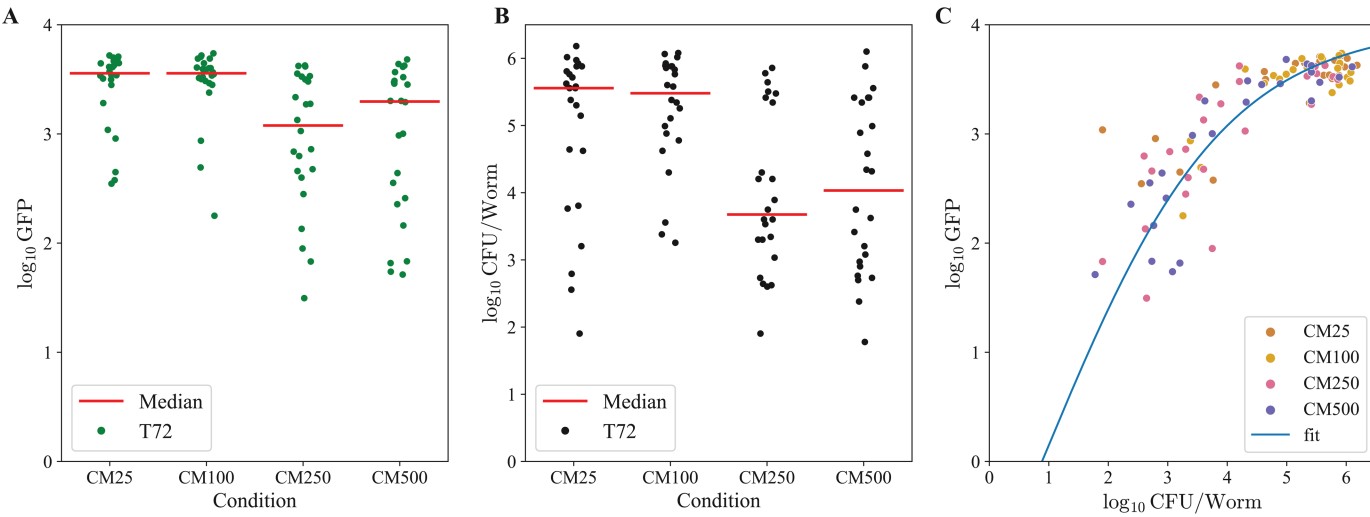

**Fig 7. Comparison of GFP and CFU distribution.** A) Green fluorescence in worms ($n = 24$) measured in BioSorter at different inhibition conditions. B) CFU/worm for the same set of worms measured through destructive sampling in all four inhibition conditions. Medians (red) for each condition for both Green and CFU/worm are also show. C) Green vs CFU/worm mapping for each worm colored by inhibition condition.

concentrations, and saturates near the device limits:

$$y(x) = a \cdot \tanh((x - b)/c), \tag{6}$$

and best fit parameters were found to be $a = 4.00$, $b = 0.884$, and $c = 3.064$.

In our models, all fits are done in the log(CFU) space, where we expect the relationship between the true bacterial load and the measurement to be more linear. Thus we also need an inverse transformation of Eq 6, which is:

$$x(y) = \frac{c}{2} \ln\left(\frac{1 + y/a}{1 - y/a}\right) + b, \tag{7}$$

where $a$, $b$, and $c$ are the same parameters fitted earlier.

When transforming distributions between the log(CFU), where all fitting is done, and the log(GFP) space, this relationship is equivalent to the variables change:

$$P(y) = P(x(y))\left|\frac{\partial x}{\partial y}\right| = P\left(\frac{c}{2} \ln\left(\frac{1 + y/a}{1 - y/a}\right) + b\right)\frac{c/a}{1 - (y/a)^2}, \tag{8}$$

where $P(x)$ is a distribution fitted in the log(CFU) space and $P(y)$ is the resulting distribution in the log(GFP) space. Example fits of a GMM of two components are shown log(CFU) and log(GFP) spaces in S1 Text: *Sample fits of models of bacterial load to GFP measurements.*

## Overview of simulations and computational methods

The mean-field simulations (dashed black lines) in Figs 1 and 2 were performed using Python's SciPy odeint solver [44]. The odeint solver was used to numerically integrate Eq 2. Best fit parameters were found using SciPy's curve_fit function to fit the log transformed

simulations to log transformed data (Table A in S1 Text). The fits were done in the log transformed space since the fluctuations were empirically more symmetric in the log space as compared to the real space. The chemical reaction like equations of the stochastic model (Eq 1) were simulated using the Gillespie algorithm [24,25]. The simulations were coded in Python and were initialized using the best fit parameters from the mean field model. Thirty stochastic Gillespie simulations were performed per condition in Fig 1. The stochastic simulations in Fig 2 ($n$ = 24, 10 repetitions) were initiated using the birth and death parameters from Table A in S1 Text, but using a colonization rate $c$ that is scaled to the bacterial density outside the worm (scale factors: 1/10, 1, 10), and carrying capacity randomly picked from 48 hour data of the highest bacterial density condition. For stochastic simulations, error bars in zero occupancy at each time point are found by calculating means and standard deviations for zero occupancies across 10 repetitions. For experimental data, the error bars are calculations using binomial standard deviation ($\sqrt{N \cdot p \cdot (1-p)}$, where p is the fraction of zeros at that time point and $N$ = 24). These are then expressed in percentages (Fig 2) for convenience.

In Fig 3, we performed a grid search for best fit parameters for variable colonization and birth rates. We found the best fit by minimizing the negative log-likelihood of the data. We evaluated the log-likelihood of the data given a model by discretizing both the experimental data and the simulation data into three levels of bacterial load: low (0-$10^{1.5}$ for MYb71, 0-10 for MYb120), medium ($10^{1.5}$–$10^4$ for MYb71, 10–$10^{2.5}$ for MYb120), and high (above $10^4$ for MYb71, above $10^{2.5}$ for MYb120) and calculating their relative probabilities. The negative log-likelihood is $\mathcal{L} = -\sum_{ij} n_{ij} \log(P_{i,j})$, where $n_{i,j}$ is the number of observed worms with bacteria at level $i$ at time $j$, and $P_{i,j}$ is the corresponding probability within the model as measured using Gillespie simulations with $n$ = 24 worms. We performed these simulations 50 times to produce the distribution of log-likelihoods shown in Fig 3B and 3D.

A Gaussian Mixture Model(GMM) with two components was fit to log(GFP) data transformed into log(CFU) data in Figs 4 and 5 using the calibrated relation from Materials and methods: Calibration of log(CFU) to log(GFP) mapping. Python's sklearn package [45] was used for fitting. Once fitted in the log(CFU) space, the distributions were then transformed back to log(GFP) as in Materials and methods: Calibration of log(CFU) to log(GFP) mapping. The best fit parameters of the mixture model are reported in tables under each panel.

For Fig 6, the stochastic differential equation of the potential model (Eq 4) was simulated in Python using the Euler-Maruyama method with a time step of 0.01 (see S1 Text: *Modeling in the log(CFU) space results in a multiplicative noise process*). Similarly, the switching model Eq 5 was also simulated in Python using the Euler-Maruyama method along with a random state switching given by the transition rates $\alpha_H$ and $\alpha_L$. Parameters for these simulations were set as described in Materials and methods: Probability distributions in the state switching model.

## Probability distributions of bacterial load in different mathematical models

Bacteria populations in both the explicit state switching model and the potential model have transitions between two distinct states, visible in these data as modes in the distribution of the total bacterial load. For each class of models, a master equation describes the probability of a system being in the high or low bacterial load state as well as the transition rates between states. For the explicit state switching model, these transition rates are parameters of the model. In contrast, in the potential model, the transition rates between states is given by properties of the potential of each state. Here we describe how these distributions are obtained for both models.

**Probability distributions in the state switching model.** In the simplest case of the state switching model, transition rates are assumed to be constant. Given a set of transition rates between states $s_{\text{low}} \xrightarrow{\alpha_H} s_{\text{high}}$ and $s_{\text{high}} \xrightarrow{\alpha_L} s_{\text{low}}$ the corresponding master equation for the probabilities of being in the high $P(s_H)$ and low state $P(s_L)$ are:

$$\frac{\partial P(s_H)}{\partial t} = \alpha_H P(s_L) - \alpha_L P(s_H) \tag{9}$$

$$\frac{\partial P(s_L)}{\partial t} = -\alpha_H P(s_L) + \alpha_L P(s_H) \tag{10}$$

Assuming that an individual worm must be in one of the two states at any given time, $P(s_H) + P(s_L) = 1$, the probability of being in the low state can be expressed as the complement of being in the high state: $P(s_L) = 1 - P(s_H)$. Using this together with Eq 9 one can solve for the probability of being in the high state as a function of time.

$$\frac{\partial P(s_H)}{\partial t} = \alpha_H - (\alpha_H + \alpha_L)P(s_H). \tag{11}$$

This is solved as:

$$P(s_H, t) = Ae^{-(\alpha_H + \alpha_L)t} + \frac{\alpha_H}{\alpha_H + \alpha_L}, \tag{12}$$

resulting in steady state values $P(s_H) = \frac{\alpha_H}{\alpha_H + \alpha_L}$ and $P(s_H) = \frac{\alpha_L}{\alpha_H + \alpha_L}$. Best fit results of Eq 12 to the weights of a GMM fit to MYb14-GFP-KmR precolonized worms at low inhibition result in transition rates $\alpha_H = 0.082 (1/\text{h})$ and $\alpha_L = 0.025\ (1/\text{h})$ (Figs 4 and 8).

We can use these results together with the stationary probability distributions of each state separately to predict the stationary probability distribution of the the full ensemble. In this model, states can be considered in isolation as they are uncoupled except through transitions. The stationary distribution for an individual state in the absence of the other is

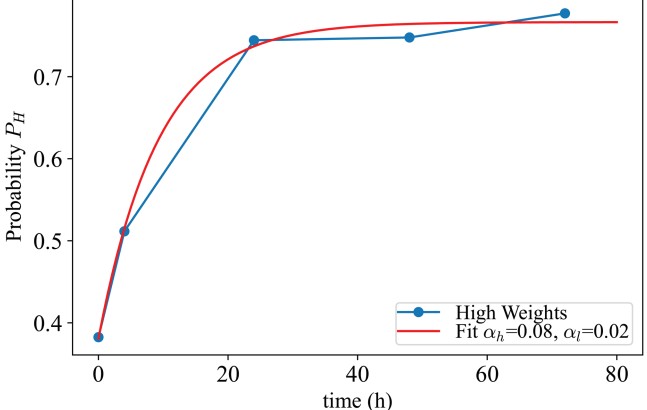

**Fig 8. Weights of the high mode in a Gaussian mixture model fit to the Green fluorescence transformed CFU data in MYb14-GFP-KmR precolonized worms over 72 hours of at low inhibition concentrations of chloramphenicol (Fig 4).** The weights serve as a proxy for $P(s_H)$ and a fit of Eq 12 results in transition rates $\alpha_H = 0.08\ (1/\text{h})$ and $\alpha_L = 0.02\ (1/\text{h})$.

$A_s \exp(-U_s(\phi)/D(s))$, where $A_s$ is a normalization constant and $U_s(\phi) = -r_s\phi^2(C_s/2 - \phi/3)$ is a potential corresponding to the force $-\frac{\partial}{\partial\phi} U_s(\phi) = r_s\phi(C - \phi)$.

The full stationary probability distribution can be thus approximated as:

$$P(\phi) = \frac{\alpha_H A_H}{\alpha_H + \alpha_L} \exp(r_H\phi^2(C_H/2 - \phi/3)/D_H) + \frac{\alpha_L A_L}{\alpha_H + \alpha_L} \exp(r_L\phi^2(C_L/2 - \phi/3)/D_L), \quad (13)$$

as long as transitions between states happens very fast compared to the other timescales in this problem.

**Probability distributions in the potential model.** There exists a set of well known results from statistical mechanics for the transition rate of a particle from a potential well to another over an energy barrier. This problem is known as Kramer's [46], problem and it obeys an Arrhenius like equation:

$$\alpha_L = \frac{\sqrt{K_H K_M}}{2\pi} \exp\left(\frac{-(U(C_M) - U(C_H))}{D}\right), \quad (14)$$

$$\alpha_H = \frac{\sqrt{K_L K_M}}{2\pi} \exp\left(\frac{-(U(C_M) - U(C_L))}{D}\right), \quad (15)$$

where $K_L = |U''(C_L)| = |f'(C_L)| \approx \frac{D}{\sigma_L^2}$, $K_M = |U''(C_M)| = |f'(C_M)|$, and $K_H = |U''(C_H)| = |f'(C_H)| \approx \frac{D}{\sigma_L^2}$. $C_L$ is the lower stable state, $C_H$ is the higher stable state, and $C_M$ is the location of the peak between these two states. Additionally, at equilibrium the stationary distributions should have a form:

$$P(\phi) = A \exp\left(\frac{-U(\phi)}{D}\right). \quad (16)$$

One feature of Kramer's result is that the switching rates depend on the variance, curvature, values of the peaks and troughs, and other features of the potential. In the switching model, these rates could in theory be independently set and be independent of the variance around each fixed point.

We can fit models for the potential to an effective potential determined directly from data

$$\frac{U_{\text{eff}}(\phi)}{D} = R(\phi) = -\ln(P(\phi)), \quad (17)$$

where $P(\phi)$ is a histogram approximation of the probability distribution of the log transformed CFU data. In Fig 9, best fits are found for a 6th order potential and a 4th order potential. The GMM can also be viewed as generating its own effective potential as the negative log of the mixture distribution. For comparison, this effective potential its corresponding force are also plotted.

It is possible to further solve for $D$ in terms of the ratio of the effective potential to $D$, denoted via $R$, its second derivative, and the measured transition rates from Eq 14:

$$D = \frac{\alpha_L 2\pi}{\sqrt{R''(C_H)R''(C_M)}} \exp\left(R(C_M) - R(C_H)\right). \quad (18)$$

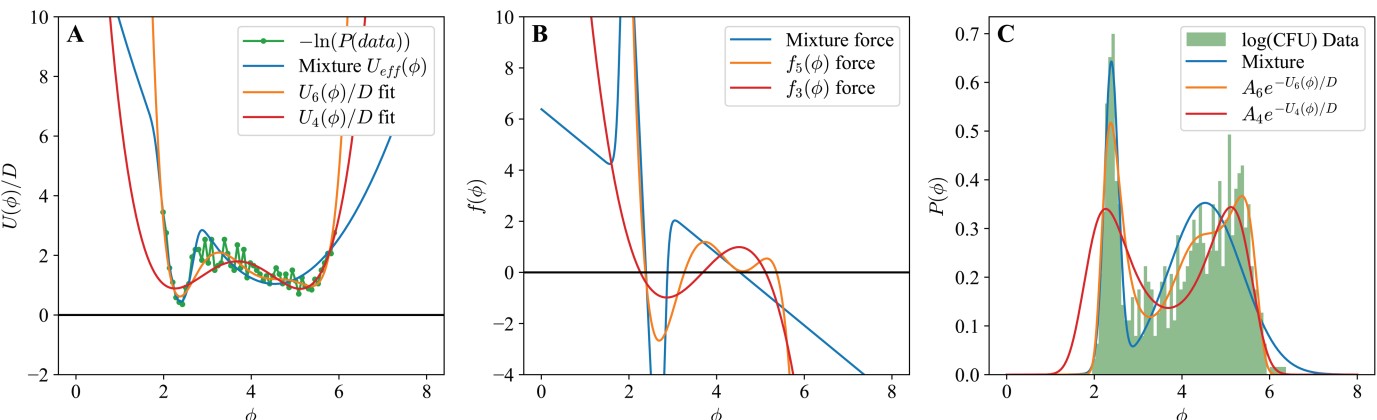

**Fig 9. Effective potentials and forces of different models.** A) Fits to the effective potential $-\log(P(\phi))$ for the GMM, a 6th order polynomial potential $U_6(\phi)$ and a 4th order potential $U_4(\phi)$. B) The corresponding force functions for the GMM, the 6th order potential ($f_5$ 5th order force), and the 4th order potential ($f_3$ 3rd order force). C) The corresponding fits of the probability distributions for all three models.

## Supporting information

**S1 Text. Figs A–H and Tables A–C.**

**Fig A in S1 Text. Bacterial load in individual worms over time.** Data are shown for bacterial species: A) MYb27 (red), B) MYb45 (green), C) MYb53 (blue), D) MYb238 (brown) to illustrate colonization dynamics. Gillespie simulations (blue lines, n=30) were carried out using parameters obtained from fitting CFU per worm data to the mean field model. Mean CFU at each time point (large blue point) and the mean-field deterministic solution (dashed black line) are also shown.

**Fig B in S1 Text. Fluorescent bead accumulation in adult worms.** Worms were exposed to high (20 $\mu$L beads per 1 mL buffer) or low (2 $\mu$L per 1 mL buffer) concentrations of beads (top labels). Additional nutrient source (heat-killed OP50 or none) is shown on the vertical labels. Samples were taken 10-60 minutes after exposure to beads, and total green fluorescence was measured on BioSorter. Each point represents one individual worm (30-60 worms per sample). Horizontal lines represent 90th percentile GFP in bead-free worms (red, HKOP50; black, no nutrient source) To the extent that the variability in the bead accumulation for the same conditions spans only about an order of magnitude, the intestinal input-output processes cannot account for a large fraction of bacterial load variability in the worms.

**Fig C in S1 Text. Bacterial load distributions under additional levels of inhibition of growth within hosts.** Green fluorescence in MYb14-GFP-KmR pre-colonized worms over 72 hours of inhibition at (a) 100$\mu$g/ml or (b) 250$\mu$g/ml concentrations of chloramphenicol (CM). Centers and weights of high and low GFP modes at each time point from the transformed GMM fits are shown in the legend. Mean GFP (black dots, dashed lines) for the entire population at each time point is shown. Mean autofluorescence (light green, dashed lines) calculated using the data from uncolonized worms is also shown. (c) Probability density fits modeled as a Gaussian mixture with two components fit on CFU transformed data and then transformed back to GFP, at T72 for CM100 and CM250 inhibition.

**Fig D in S1 Text. Bacterial load switching between modes in an additional experimental run.** Green fluorescence over 48 hours after MYb14-GFP-KmR pre-colonized worms were separated based on high (A,E) and low (B,F) GFP under conditions of A-C) No Migration

(top) or E-F) Migration (bottom). Mean GFP (black dots, dashed lines) for the entire population at each time point is shown. Mean auto-fluorescence (light green, dashed lines) calculated using the data from uncolonized worms is also shown. C,F) PDFs using transformed GMM fits at T48 for worms starting at low (light green) and high (dark green) fluorescence in no migration (top) and migration (bottom) conditions.

**Fig E in S1 Text. State switching in N2 adults colonized with *Salmonella enterica* LT2.** Adult worms were allowed to feed on lawns of *S. enterica*-GFP for 48 hours, then purged and sorted into bins based on GFP fluorescence as before. Individual worms were distributed into wells of a 96-well plate containing S medium + 1X heat-killed OP50 + 50 $\mu$g/mL each kanamycin (for selection) and chloramphenicol (to prevent re-inoculation) for 48 hours. Individual worms ($n = 24$) were digested at the start (time 0) and end (time 48) of the experiment.

**Fig F in S1 Text. Probability distributions of GFP and CFU data.** Left: log(CFU) probability distribution using transformed log(GFP) from Fig 4 at 24 hours. GMM with two components is fitted to the log(CFU) data. Right: The GMM distribution fitted in the log(CFU) space is transformed back to the log(GFP) space.

**Fig G in S1 Text. Predictions of dynamics in the 6th order polynomial potential (5th order force) model.** We fit the potential to data in Fig 4 at 24 hours. We start then with the initial distribution of bacteria from the High GFP condition of Fig 5 (top row) (shown as a histogram in the top left panel here), and fit it to a GMM (orange line in the top left panel). We evolve this distribution according to the dynamics in Materials and methods: Probability distributions in the potential model (histograms in all other panels), and compare to the GMM fits to 24 and 48 hour data in Fig 5 (shown here as orange lines in the last two panels). Times 6,12, and 18 are from simulation alone are for visualization purposes only, we do not have the corresponding data for these times.

**Fig H in S1 Text.Predictions of dynamics in the state switching model**. Plotting conventions are the same as in Fig G in S1 Text, except that the fits and the dynamics are now done using the state switching model. Times 6,12, and 18 are from simulation alone are for visualization purposes only, we do not have the corresponding data for these times.

**Table A in S1 Text. Parameters from logistic model fits to bacterial load data during mono-colonization of N2 worms** (Fig 1, Fig A in S1 Text).

**Table B in S1 Text. Raw parameters from the GMM in the log(CFU) space fitted to data in Fig 4.**

**Table C in S1 Text.Raw parameters from the GMM in the log(CFU) space fitted to data in Fig 5.**
(PDF)

## Acknowledgments

We thank Daniel Weissman for productive discussions on the manuscript.

## Author contributions

**Conceptualization:** Satya Spandana Boddu, K. Michael Martini, Ilya Nemenman, Nic M. Vega.

**Data curation:** Satya Spandana Boddu, K. Michael Martini, Nic M. Vega.

**Formal analysis:** Satya Spandana Boddu, K. Michael Martini, Nic M. Vega.

**Funding acquisition:** Ilya Nemenman, Nic M. Vega.

**Investigation:** Satya Spandana Boddu, Nic M. Vega.

**Methodology:** Satya Spandana Boddu, K. Michael Martini, Ilya Nemenman, Nic M. Vega.

**Project administration:** Satya Spandana Boddu, K. Michael Martini, Ilya Nemenman, Nic M. Vega.

**Resources:** Ilya Nemenman, Nic M. Vega.

**Software:** Satya Spandana Boddu, K. Michael Martini, Nic M. Vega.

**Supervision:** K. Michael Martini, Ilya Nemenman, Nic M. Vega.

**Validation:** Satya Spandana Boddu, K. Michael Martini, Nic M. Vega.

**Visualization:** Satya Spandana Boddu, K. Michael Martini, Nic M. Vega.

**Writing – original draft:** Satya Spandana Boddu, K. Michael Martini, Ilya Nemenman, Nic M. Vega.

**Writing – review & editing:** Satya Spandana Boddu, K. Michael Martini, Ilya Nemenman, Nic M. Vega.

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
