## [Decision Letter · Decision Letter 0]

20 Mar 2025

PCOMPBIOL-D-25-00094

Variance in C. elegans gut bacterial load suggests complex host-microbe dynamics

PLOS Computational Biology

Dear Dr. Boddu,

Thank you for submitting your manuscript to PLOS Computational Biology. After careful consideration, we feel that it has merit but does not fully meet PLOS Computational Biology's publication criteria as it currently stands. Therefore, we invite you to submit a revised version of the manuscript that addresses the points raised during the review process.

Please submit your revised manuscript within 30 days May 20 2025 11:59PM. If you will need more time than this to complete your revisions, please reply to this message or contact the journal office at ploscompbiol@plos.org. Please include the following items when submitting your revised manuscript:

We look forward to receiving your revised manuscript.

Kind regards,

Samir Suweis, Ph.D.

Academic Editor

PLOS Computational Biology

Natalia Komarova

Section Editor

PLOS Computational Biology

**Additional Editor Comments:**

Both reviewers find your work very valuable and well written. They propose some suggestions and comments that I invite the authors to consider to further improve their work.

**Journal Requirements:**

At this stage, the following Authors/Authors require contributions: Satya Spandana Boddu, K. Michael Martini, Ilya Nemenman, and Nic Vega. Please ensure that the full contributions of each author are acknowledged in the "Add/Edit/Remove Authors" section of our submission form.

4) We notice that your supplementary Figures, text, and Tables are included in the manuscript file. Please remove them and upload them with the file type 'Supporting Information'. Please ensure that each Supporting Information file has a legend listed in the manuscript after the references list. Please cite and label the supplementary figures and tables as "S1 Figure", "S2 Figure", “S1 Table” and “S2 Table,” and so forth.  

Note: Please ensure that the funders and grant numbers match between the Financial Disclosure field and the Funding Information tab in your submission form. The funders must be provided in the same order in both places as well.

**Reviewers' comments:**

Reviewer's Responses to Questions

Reviewer #1: The study investigates variability in C. elegans gut bacterial load and proposes that this variability cannot be explained by demographic noise or static host heterogeneity alone. Through a combination of experiments and mathematical modeling, the authors suggest that the host-microbe system switches stochastically between high- and low-growth phenotypes.

The article is quite clear and presents results that, to the best of my knowledge, are original and are undoubtedly on par with Plos Computational Biology. I have questions about the part about the model, particularly with respect to the mechanisms that were decided to be incorporated.

1) Why is a white noise model chosen instead of considering a colored noise process or state-dependent fluctuations? Is it merely a decision of convenience, or is there a clear biological motivation?

2) It appears that the authors argue that the observed variability in bacterial load within C. elegans does not require microbial interactions to explain, as even when the worms were mono-colonized (i.e., exposed to a single bacterial species), a large variability in bacterial load was still observed. However, they do not delve into possible mechanisms by which the absence of microbial interactions is not a relevant factor. For example, they do not rule out that previous microbial interactions or environmental factors may have influenced the observed dynamics. Similarly, they do not explore whether the variability could be better explained in a context of competition or cooperation between bacteria rather than just considering stochastic state transitions. Was the possibility of past or indirect microbial interactions influencing the observed variability evaluated, even under mono-colonization conditions? Were models of competition or cooperation between bacteria explicitly ruled out?

Once the authors addressed, at least qualitatively, the two issue above, the manuscript will be suitable for pubblication in Plos Computational Biology.

Reviewer #2: This paper addresses the important question of the in vivo colonization of the gut by microbes. The approach taken of focusing of one bacterial species in a host which presents as best as possible reproducible conditions is very relevant to understand this complex problem. The authors showed in an extensive manner that the logistic model could not explained their data. They propose two models that can reproduce the data and suggest experiments which should be able to discriminate between the two models. I recommend this paper for publication.

Comments

- Line 78: Could you explain why what would be expected for \sigma/\mu from demographic noise? It does not seem obvious to me from [23] that the linearity between mean and standard deviation exclude the demographic noise as the source of variability.

- Line 81: Could you be more specific about the long-tailed distribution and fit your experimental data with the specific distribution which is expected?

- Very nice Figure 5!

- For model 1, could you explain the shape of $f(\phi)$ with biological arguments?

- For model 2, it would be interesting to propose hypotheses as to why they are two states. What would be the biological origin of those two different states? What would trigger the change from one to the other state?

- Could it help discriminating between the model to add “fit data” at intermediates times (6 hours, 12 hours, 18hours) in Fig 16 and 17? From Fig1 for instance it seems you do have access to experimental data to do so.

- Page 15: Line 454: Clearly, as the order of the potential model is increased it becomes better able to fit the empirical bacterial load distribution.

Could you comment on the fact that you are not actually overfitting? (also for the state switching model)

Minor comments

Page 1: In the author’s summary, I think the sentence is difficult to understand (especially before having read the paper): We found that bacteria behaved differently when grown in a host compared to the standard logistic growth observed in vitro, exhibiting population density-dependent growth or the emergence of two primary subpopulations of bacteria. I would reformulate.

Page 2: We found that high variation is still present in bacterial load of mono-colonized worms, indicating that microbe-microbe interactions are not required to produce inter-individual differences in microbial colonization of the worm intestine.

This sentence suggests that microbe-microbe interactions will produce inter-individual differences. Is it possible that those interactions will decrease them? (by constraining the dynamical behavior for instance, or some sort of self-organisation).

Page 2: Instead, we found emergence of apparent alternative states in bacterial load inside worms, with movement of individuals between these states, suggesting that canonical models of bacterial growth cannot fully characterize the host-microbe system’s dynamics.

I would use the word transitions rather than movement.

Page 2 last paragraph: I find it confusing that you write that the two models cannot be distinguished based on snapshots data while you do have time series available.

Page 4: In Fig1 B (and C), how could the deterministic curve to on top of simulations? The deterministic curve should be the average of all simulations. Maybe my impression from the figure is wrong because of the log scale. The figures B and C are convincing, but it could be easier to have violin plots at the time points where the experimental data is taken for the theoretical model to compare the variance in the experimental and theoretical cases. (Maybe around t=10h for instance, the variance is well captured by the model?)

Page 5: to help the reader, I would add the name of the parameters after ingestion and excretion rates on line 117. Is it correct that those parameters are named as follow: bi = birth parameter, ci colonization/ingestion parameter, di excretion/death?

Page 16: You could cite https://doi.org/10.7554/eLife.55650 with ref [39]. The papers reached the same conclusions with different approaches.

Page 24: Fig. 10: it is not obvious to me that the fits are good (in particular for panel C). My impression is maybe due to the log scale. Could you provide a quantitative assessment of the quality of the fit?

Page 31-32: could be nice to combine figures 16 and 17 to be able to compare more easily.

Rem: The supporting information is not well structured. Could you remove the “1” in the title page 27 and “2” page 28? Or maybe start with a “1” on page 24 for “Logistic growth model fits …” ?

**Have the authors made all data and (if applicable) computational code underlying the findings in their manuscript fully available?**

Reviewer #1: Yes

Reviewer #2: Yes

PLOS authors have the option to publish the peer review history of their article (what does this mean?). If published, this will include your full peer review and any attached files.

Reviewer #1: No

Reviewer #2: No

**Figure resubmission:**
---

## [Editor Report · Decision Letter 1]

2 May 2025

Dear Dr. Boddu,

We are pleased to inform you that your manuscript 'Variance in C. elegans gut bacterial load suggests complex host-microbe dynamics' has been provisionally accepted for publication in PLOS Computational Biology.

Best regards,

Samir Suweis, Ph.D.

Academic Editor

PLOS Computational Biology

Natalia Komarova

Section Editor

PLOS Computational Biology

---

## [Editor Report · Acceptance letter]

PCOMPBIOL-D-25-00094R1

Variance in *C. elegans* gut bacterial load suggests complex host-microbe dynamics

Dear Dr Boddu,

I am pleased to inform you that your manuscript has been formally accepted for publication in PLOS Computational Biology. Your manuscript is now with our production department and you will be notified of the publication date in due course.

With kind regards,

Zsofia Freund
